

# A Dual-Droplet Approach for Measuring the Hygroscopicity of Aqueous Aerosol

Jack M. Choczynski,[1] Ravleen Kaur Kohli,[1] Craig S. Sheldon,[1] Chelsea L. Price,[1] James F. Davies[1]*

[1]*Department of Chemistry, University of California Riverside, Riverside CA 92521 USA*

*Correspondence to:* James F. Davies (jfdavies@ucr.edu)

**Abstract.** Accurate characterization of the water activity and hygroscopicity of aqueous aerosol material allows us to predict the chemical and physical state of aerosol particles exposed to humid conditions in the environment. The

hygroscopicity of aerosol determines the size, phase morphology, viscosity, chemical reactivity, and optical properties of constituent particles, and directly impacts their ability to form clouds in the atmosphere. In this work, we describe measurements of hygroscopicity using a linear quadrupole electrodynamic balance (LQ-EDB). We levitate two droplets, one droplet that acts as a relative humidity (RH) probe and one sample droplet, and expose them to controlled environmental conditions. We describe the development of a RH measurement using probe droplets of aqueous NaCl

or LiCl, allowing for precise *in-situ* measurements of RH in the LQ-EDB chamber. We demonstrate that the RH may be determined with an accuracy of 0.5% at 50% RH and better than 0.1% at 90% RH using NaCl, and show that LiCl is effective at characterizing the RH from ~10% RH up to ~90%. We simultaneously measure the response of sample droplets containing aqueous material (including ammonium sulfate, citric acid, 1,2,6-hexanetriol, and tetraethylene glycol) and report hygroscopic growth via their radial growth factors. We use established thermodynamic models to

validate the accuracy of the RH probe and to compare with the measured hygroscopicity of the samples. This approach shows significant advantages over other methods for accurately characterizing the hygroscopicity of samples with a range of characteristics, such as high viscosity and vapor pressure.

## 1 Introduction

The hygroscopicity of aerosol has been of interest to scientists for many years as it plays an important role in the phase

morphology, chemical reactivity, optical properties and cloud forming efficiency of aerosol particles (Chim et al., 2017; Freedman, 2017; Riemer et al., 2019; Chan et al., 2005; Lam et al., 2020; Denjean et al., 2015; Kulmala et al., 2017). Hygroscopicity describes the ability of a solute to lower the water activity in solution – at the same molar concentration, a hygroscopic compound will reduce the water activity to a greater extent than a non-hygroscopic or less hygroscopic compound. For ideal aqueous solutions, Raoult's law may be used to equate the water activity to the

mole fraction of water. However, aqueous aerosol in the atmosphere are rarely ideal solutions, and relating the water activity to the amount of solute is more complicated. For aerosol at equilibrium, the water activity is determined by the water saturation ratio ($S_{env}$) of the gas phase:

$$S_{env} = \frac{p_w(T)}{p_w^\circ(T)} = \frac{RH}{100}, \tag{1}$$



where $p_w(T)$ is the vapor pressure of water at temperature T, and $p_w^\circ(T)$ is the saturation vapor pressure. The relative

humidity (RH) is equal to this fraction expressed as a percentage. The hygroscopicity may be equivalently described as the amount of condensed phase water associated with aerosol under a given RH and will determine many physical and chemical characteristics of the aerosol that vary with the amount of water, including viscosity (Reid et al., 2018), refractive index (Cotterell et al., 2017; Day et al., 2000), rates of molecular diffusion (Bones et al., 2012; Marshall et al., 2016; Shiraiwa et al., 2011), reactive uptake coefficients (Davies and Wilson, 2015; Grvzinic et al., 2015), and

acidity and pH (Pye et al., 2020). Additionally, hygroscopicity plays an important role in regulating cloud formation via the Köhler equation. Aerosol particles capable of forming cloud droplets, known as cloud condensation nuclei (CCN), exhibit a water saturation ratio ($S_d$) that is a function of the water activity of the aqueous phase and the Kelvin factor. At equilibrium, $S_{env} = S_d$, but for cloud formation to occur, $S_{env}$ must rise above the maximum value of $S_d$, known as the critical supersaturation, a value that is reduced by hygroscopic solutes. While hygroscopicity and CCN

activity are closely related, studies have shown that measurements of hygroscopicity at subsaturated RH can deviate from estimates of hygroscopicity derived from CCN activity measurements, attributable to solubility and surface tension effects (Petters and Kreidenweis, 2013, 2008, 2007; Ruehl et al., 2010, 2016). Deriving a complete understanding of the CCN activity and physical properties of aerosol particles across the full range of RH requires a careful approach utilizing both experimental methods and models (Tang et al., 2019).


To characterize aerosol hygroscopicity, it is common to measure the hygroscopic growth of particles as a function of RH in subsaturated conditions, necessitating accurate measurements of particle size and RH. Given that size depends on the amount of non-volatile hygroscopic material, hygroscopic growth measurements are typically reported using growth factors, relating the hydrated size at a given RH to the dry amount of solute at 0% RH. Determining the RH

using capacitance probes is common, however most electrical probes suffer from uncertainties on the order $1 - 2\%$ and are typically positioned *ex-situ* with respect to the sample (Chen and Chen, 2017). There are several methods that use the response of a probe droplet to determine the *in-situ* RH with very high accuracy and precision. The first example of this was reported by Walker et al. using an optical tweezers capable of confining multiple droplets (Walker et al., 2010). One droplet would contain a well-understood solute, such as sodium chloride, to act as a probe, while

the other droplet would contain the sample of interest, such as ammonium sulfate. The equilibrium response of the probe droplet would inform the RH of the system, allowing the response of the sample to be attributed to an RH with sub-1% accuracy. While effective, this method proved challenging as confining droplets with independent compositions is not straightforward in an optical tweezers, as samples are introduced in a nebulized aerosol plume, and some assumptions of the initial state of the particles is required that can lead to some uncertainty in the

measurement. Furthermore, the hygroscopicity of semi-volatile material can be challenging to measure using equilibrium droplet approaches, as the amount of solute changes over the course of the measurement. The more rapid the measurement, the less significant the change in dry mass becomes. For typical hygroscopic growth measurements, the RH is changed slowly to allow for the probes and sample to fully equilibrate.



More recently, a hygroscopic growth method was developed using a cylindrical electrodynamic balance where the rate of water evaporation was used as a measure of the RH (Davies et al., 2013). Sample droplets and probe droplets were introduced sequentially, and the rate of water evaporation from the sample droplet was used to infer the water activity in the droplet, with its size giving an indication of the solute concentration. This method infers equilibrium hygroscopicity from water evaporation kinetics and has proven to be rapid and accurate in characterizing the

hygroscopic growth at high RH, with some limitations (Rovelli et al., 2016). The rate of water evaporation cannot be too fast, as this leads to temperature changes in the droplet that are not accurately captured by mass transport models (Su et al., 2018). Furthermore, water transport kinetics are assumed to be limited only by gas-phase diffusion, and thus water diffusion limitations in the droplet will lead to increased uncertainty. Slow evaporation and rapid water diffusion are typically encountered at high RH, but at low RH diffusion limitations in the particle may limit the rate of water

evaporation, leading to the erroneous assumption that the solute lowers the water activity and is more hygroscopic.

In this work, we present an updated equilibrium hygroscopicity measurement using multiple droplets confined within a linear-quadrupole EDB (LQ-EDB). We describe the new method and show hygroscopic growth data compared to literature data and thermodynamic models. We demonstrate several advantages over previous methods: (1) The sample

and probe droplets can be introduced independently, with no risk of cross-contamination. (2) The equilibrium size is measured allowing the hygroscopicity of samples that exhibit kinetic limitations to be studied. (3) Rapid RH growth curves can be obtained allowing semi-volatile species to be probed. (4) RH is determined *in-situ* with high accuracy across the whole RH range.

## 2 Methods

### 2.1 Particle Levitation

A linear quadrupole electrodynamic balance (LQ-EDB) was used to levitate multiple droplets in a linear array under controlled environmental conditions. The general technique and our specific implementation has been described previously (Hart et al., 2015; Jacobs et al., 2017; Price et al., 2020; Davies, 2019). Briefly, droplets are introduced into an aluminum chamber using a piezoelectric droplet dispenser (Microfab MJ-ABP-01) powered with an in-house

developed voltage pulser. As the droplets are produced, a net negative charge is imparted due to the presence of an induction electrode set at +200 to +500 V and producing a charge on the order of 20 – 100 fC (Davies, 2019; Haddrell et al., 2012). The droplets become confined to the linear axis of the rods and fall to an equilibrium height above a balancing DC electrode at voltage $V_{DC}$. In this work, two separate droplet dispensers and induction electrodes are used to introduce two droplets of different composition in quick succession. This allows both droplets to be stabilized by

the electric fields and their position in the trap is varied with a single $V_{DC}$, regulated by a PID feedback loop to stabilize the position. Both droplets are illuminated by a laser for visual analysis, and one droplet at a time is illuminated by an LED for spectroscopic analysis, with full manual and automated control through a LabVIEW interface.





### 2.2 Optical Characterization

When illuminated by broadband light from an LED, droplets will act as a spherical cavity and certain wavelengths
will meet the condition for total internal reflection and become resonant in the cavity. Light of these wavelength is
scattered efficiently and will appear as sharp peaks in the backscattered spectrum. We measure the backscattered
spectrum using an Ocean Optics HR4000+ spectrometer and observe clear resonant peaks – this is the Mie resonance
spectrum (see Figure S1). Using Mie theory, the algorithms of Preston et al. (Preston and Reid, 2013), and an in-house
developed LabVIEW user interface, the position of the peaks are determined and used to find the radius and the real
part of the refractive index (RI) using a first-order Cauchy dispersion expression, with parameters $m_0$ and $m_1$. The
diameter is determined with ~10 nm accuracy while the refractive index uncertainty ranges from ±0.001 to ±0.005
depending on the peaks present in the spectrum. To verify the resulting size and RI, a full spectrum is simulated by
Mie theory using the best-fit size and RI and used for visual verification of the fit, as shown in Figure S1. The spectra
are measured over the wavelength range 640-680 nm and the RI is reported at 589 nm using the 1st order Cauchy
relation:

$$RI(\lambda) = m_0 + \frac{m_1}{\lambda^2}, \tag{2}$$

### 2.3 Multiple Droplet Method

In order to characterize the size and RI of multiple droplets, we programmatically move the droplets up and down by
varying $V_{DC}$ to switch the sample illuminated by the LED. In order to ensure that any changes in position are
compensated for by the DC feedback loop, the droplets are switched every 10 s. This leads to some blurred spectra
during the switch between both samples, however these do not impact the analysis of the other frames and can be
easily discarded. Spectra are saved in separate files for each sample and analyzed as described above.

### 2.4 Characterizing the Dry Size

In order to characterize the hygroscopicity, a measure of the amount of solute is required. Usually, this is referenced
to the size of the dry particle, either in terms of the dry mass or a spherical equivalent dry volume, and mass or radial
growth factors (mGF or rGF) are reported. However, direct measurements of the dry particle size are challenging. In
a typical EDB, the balancing voltage $V_0$ under dry conditions is used to indicate the dry mass, although this has some
limitations as it assumes a constant charge over time, which is not always achieved (Haddrell et al., 2012). Other
methods have used the starting size and solute concentration (Davies et al., 2013; Rovelli et al., 2016), or the measured
refractive index (Cotterell et al., 2014). Here, we compare several approaches for determining the dry size. The first
relies on measuring the deliquescence of well-characterized solid particles – this approach is used for inorganic
particles and is described in *Sect 3.1*. We compare this method to the use of the refractive index to characterize solute
concentration and subsequently determine dry size. For particles without a well-defined deliquescence point, such as
most organic particles, we explore two further options – one involves simply measuring the size under 0% RH, while
the other correlates the measured size at high RH with the predicted growth factor from a thermodynamic model (see
Sect 2.6*)*, allowing the dry size to be estimated. These methods are discussed further and applied in Sect 3.2.





## 2.5 Environmental Conditions

The relative humidity in the chamber is regulated by varying the mixing ratio of humidified and dry $N_2$ gas at a flow rate of 200 sccm. The water bath used to generate the humidity is housed in the reservoir of a recirculating chiller, which is used to control the temperature of the LQ-EDB chamber and avoid temperature changes associated with lab temperature deviations. The temperature of the chamber and water bath are held below ambient (at 18 °C) to reduce condensation in the tubing. This setup allows the RH to be varied from dry conditions up to ~95% RH. For the measurements described here, the RH was either varied stepwise, in 5-10% increments, or in a pseudo-continuous manner, where 0.5% RH changes were initiated in time intervals down to 5 s, allowing the full RH range to be sampled in under 20 minutes.

## 2.6 Thermodynamic Models

We make use of two common thermodynamic models in this work – the Extended Aerosol Inorganic Model (E-AIM) (Clegg et al., 1998, 2001) and the Aerosol Organic Inorganic Mixtures Functional groups Activity Coefficients (AIOMFAC) (Zuend et al., 2011, 2008) model. These both allow the RH-dependent properties of inorganic and organic particles to be predicted based on empirical data and thermodynamic predictions. The E-AIM model reports a density along with mass fraction data, allowing radial growth factors to be calculated. The AIOMFAC model only reports a mass fraction, and the density is estimated using a volume-additive approach, as described in the SI. The models are used to compare with our measurements and for deriving the RH from measured radial growth factors of probe particles.

## 3 Results and Discussion

### 3.1 *In-Situ* Measurements of Relative Humidity using a Droplet

The relative humidity inside the chamber is a function of the amount of gas phase water and temperature. The RH may be probed using a capacitance probe or a dew-point hygrometer, but these methods have limited accuracy due to their non-linear response, hysteresis on hydration/dehydration, and their location relative to the sample that may result in temperature changes. The size response of a droplet to changes in RH may be used to indicate the RH for well-characterized particles. In this section, we describe the development and use of a NaCl droplet to probe RH >50%, and LiCl to probe RH >10%. The upper RH is limited by the dynamic range of size measurement accessible to the Mie resonance sizing method.

### 3.1.1 Sodium Chloride Droplet-Based RH Probe

The size and refractive index of two sodium chloride particles was measured as the RH was varied from around 90% to 50% RH (Figure 1(a)). These data were using to infer the RH in two ways: (1 – the size method) a polynomial fit (with coefficients shown in Table 1) of water activity versus radial growth factor (GF), obtained from E-AIM (Clegg et al., 1998, 2001), was used to relate measured GF to RH; and (2 – the RI method) the measured RI was used to calculate the RH based on the parameterization of Cotterell et al. (Cotterell et al., 2017). In order to determine the radial GF, the dry size of each particle was determined from the size of the droplet at its deliquescence RH (dRH =

75.5%), identified by slowly increasing the RH exposed to a dry particle of NaCl (see *Supplemental Information* and Figure S2). A radial GF of 1.9 was used to calculate the dry size, determined from E-AIM, and the GF for all other sizes was determined. The RH determined by the size and RI methods are shown in Figure 1(b) and 1(c), respectively, for both droplets.

The size of the droplet is obtained with a high precision, within 10 nm, while the RI is less well-constrained, varying by ±0.001 to ±0.005. When determining the RH, the larger uncertainty in the RI leads to a larger uncertainty in RH, with an estimated error of ±1.0%. The size method, on the other hand, results in an RH uncertainty of ±0.25% at 50% RH and a diminishingly small uncertainty at high RH, with ±0.05% at >90% RH. The uncertainty in the size becomes

less significant at high RH, while the uncertainty in the RI becomes more significant, due to how these vary with RH (Figure S3).

The absolute accuracy of these methods depend on the accuracy of the source data. E-AIM has been developed with many experiments used to correct theoretical prediction with empirical relationships. Given the abundance of data

describing the hygroscopicity of NaCl, these predictions are considered highly accurate (Peng et al., 2016; Zieger et al., 2017; Laskina et al., 2015). In the work of Cotterell et al. (Cotterell et al., 2017), the RI is measured using optical tweezers as a function of RH. The resulting parameterization will implicitly encompass the uncertainty of the RH probes used in that work, thus negating the advantages of using a droplet as an RH probe. There is a relatively broad spread of refractive index values attributed to supersaturated NaCl solutions, and at 50% RH the RI values in the

literature span ~1.41 – 1.42, while at high RH the agreement is much closer (Bain and Preston, 2020). Due to the higher precision and accuracy of the size method compared to the RI method for determining the RH, we will adopt the size method going forwards to determine RH using a droplet probe.

To test the accuracy of the droplet probe method, a direct comparison to another well-characterized chemical system

was performed. Ammonium sulfate (AS) is the most abundant salt in the atmosphere and the most widely studied atmospherically relevant inorganic species. Using an NaCl droplet as a probe of the RH, the radius of an AS droplet was measured over the RH range from ~45% to 95%. The size of the AS droplet at the deliquescence RH (dRH = 80%) and corresponding growth factor of 1.48 was used to determine the dry size of the particle to calculate the radial growth factors. Both discrete RH steps and continuous linear RH changes were applied to samples. Figure 2 shows

the hygroscopic growth curve of AS determined in this way as compared to the E-AIM prediction, with agreement within 0.5% (at worst) until just below 50% RH, where the data deviates from the prediction. This is attributed to a breakdown of the RH measurement using NaCl as a probe below this point, as evidenced by the apparent RH showing ~40% at the point of efflorescence for NaCl, at a growth factor of ~1.52. This is lower than the typical eRH of NaCl (~45%, although not a thermodynamically well-defined point) (Gao et al., 2007). The accuracy of the NaCl probe is

therefore judged to be sufficient only above 50% RH.



### 3.1.2 Lithium Chloride Droplet-Based RH Probe

The efflorescence of NaCl at ~45% RH and inaccuracy at <50% RH limits its application to measure RH under dry conditions. Instead, LiCl, which deliquesces at 11.3% RH and effloresces only under exposure to <10% RH, may be used. Unlike NaCl, LiCl is not well-characterized or represented by thermodynamic models. Water activity data has been reported by Robinson et al. (Robinson, 1945), and the AIOMFAC model predicts a similar dependence (Zuend et al., 2011). However, translating the reported mass fraction dependence to a growth factor dependence requires knowledge of the solution density. Figure 3 shows the radial GF of LiCl derived from the mass fraction data reported by AIOMFAC. A volume additive density approached was adopted, as described in the *Supplemental Information*. The radial GF at the deliquescence point is taken to be 1.525, and measurements of LiCl using an NaCl droplet probe were performed, showing a clear deviation from the AIOMFAC predication below ~70% RH, but with agreement at higher RH within ~0.5%. A lack of available data with an NaCl probe below 50% RH was remedied using an ammonium sulfate droplet, extending the LiCl data down to ~35%. A polynomial fit of the RH was determined from the experimental data and connected to the point of deliquescence. The accuracy of the RH is expected to be limited over the range where data is not available (12% to 35% RH), but the precision and reproducibility is on the order of ±0.1% based on the uncertainty in the size measurements. The coefficients of the polynomial fit are shown in Table 1.

### 3.1.3 Advantages and Limitations of the Dual-Droplet Method

A key advantage of this approach come from how it can account for the effects of temperature, which are known to cause problems with external RH probes, but can be fully accounted for with an *in-situ* probe of RH. The solubility of NaCl varies only weakly with temperature and changes in hygroscopicity and deliquescence RH may be accounted for using different E-AIM models that account for temperature (Friese and Ebel, 2010). The accuracy and precision of this approach is also a significant advantage, with a time response for non-viscous particles limited only by the condensation or evaporation rate of water, which for droplets on the order of 10 μm in size is on the order of a few seconds (Miles et al., 2010).

Despite these advantages, there are some limitations, particularly in the dynamic range of RH that can be probed in a single measurement. Due to the need to calculate the size of the particle at dRH in order to determine accurate growth factors, the range of RH that may be probed using LiCl is limited. A particle of 3500 nm at dRH, will be ~8000 nm at 95% RH, spanning the range over which the size can be reliably determined accurately. Larger sizes may be determined, but the uncertainty increases due to the number of higher order peaks present in the Mie resonance spectrum. For a similar droplet of NaCl, the highest RH that may be reliably determined is ~99%. Less hygroscopic probes may be developed to explore systems closer to saturation, but this range is currently inaccessible to our methods and will not be explored presently.





### 3.2 Hygroscopic Growth of Organic Particles

For particles that do not show a deliquescence transition from a dry particle to a wet droplet, determining the dry size challenging. As introduced earlier, there are three options available, depending on the sample, that offer varying advantages and disadvantages. (1) The measured refractive index can be compared to a calibration of bulk solution RI as a function of concentration, or literature data, to determine the composition and infer the radial growth factor and/or mass fraction. This approach is widely applicable and is not impacted by changes in dry size due to evaporation.

However, the uncertainty associated with the RI is much larger than the radius, leading to reduced accuracy of the hygroscopic growth measurements. (2) At high RH, thermodynamic model predictions are well-validated as these coincide with solute concentrations accessible to bulk solutions. Thus, the radial growth factor can be constrained to match the thermodynamic modelling predication at high RH in order to determine a dry size. This approach is limited by the accuracy of the model and, for AIOMFAC for example, the accuracy of the density approach used to convert

from mass fraction to radial growth factor. (3) When particles are anhydrous under dry conditions, the measured size can be used to determine the dry size. This approach is effective for non-volatile components, but is limited when samples are semi-volatile and the dry size changes over time.

The three approaches described here are applied to citric acid, 1,2,6-hexanetriol, and tetra-ethylene glycol in the

subsequent sections.

### 3.2.1 Hygroscopicity of Citric Acid

Citric acid is a highly oxygenated organic molecule that has been used in several studies as a proxy for oxidized organic material in the atmosphere (Davies and Wilson, 2015; Grvzinic et al., 2015; Chim et al., 2018). It is highly soluble and will form a viscous aqueous solution at low RH (Davies and Wilson, 2016; Lienhard et al., 2014). The

high viscosity limits the rate of transport of water and can yield measurements of states that have not reached equilibrium if insufficient time has passed in a measurement. Due to the formation of a viscous state at low RH, citric acid particles do not effloresce and may not reach a fully anhydrous state even under dry conditions (Peng et al., 2001).

We measured the size of citric acid droplets across the full RH range alongside an LiCl droplet to probe the RH. At low RH, measurements were performed for extended time periods (up to hours) to ensure the sample droplet achieved an equilibrium composition. The AIOMFAC-derived growth factor was determined using a volume-additive density approach (see *Supplementary Information*). The experimental growth factor was tagged to the AIOMFAC prediction at the highest measured RH, shown in Figure 4(a) and 4(b). In the volume-additive density approach, when the crystal

density ($\rho = 1.66$ g/cm$^3$), the AIOMFAC curve deviates significantly from our data below ~70% RH (Figure 4(a)). If instead an estimate of the sub-cooled melt density is used for the density of CA in solution, as predicted using the online tool UManSysProp ($\rho = 1.47$ g/cm$^3$) (Barley et al., 2013), the AIOMFAC prediction and data agree closely, with the largest deviation of 0.4% of the radial growth factor (Figure 4(b)). Both predictions reference a dry particle with a density equal to the crystal density. It is clear these data do not tend towards unity under dry conditions and



this is due to the formation of a stable hydrate and the amorphous state of the particle – the citric acid will not form a crystal even at 0% RH and thus the size will always be larger.

An alternative method for determining the radial growth factor is via the measured refractive index (RI). Using the data of Lienhard et al. (Lienhard et al., 2012), we determine the mass fraction of solute (MFS) associated with the

measured RI and compare these directly to AIOMFAC in Figure 4(c). By estimating the density, we can also determine the radial growth factor from these MFS, and these estimates are shown in Figure 4(a) and 4(b), using the crystal and sub-cooled density, respectively. Both estimates yield radial growth that is significantly lower than the AIOMFAC predictions, and mass fractions that are higher. Indeed, at the lowest RH, a mass fraction > 1 is predicted, which is likely erroneous. To explore the origin of this error, we closely compared the Mie resonance spectra with the predicted

spectra based on the best fit size and RI, and observe excellent agreement for both high and low RH spectra (Figure 5(a) and 5(b), respectively). Equivalently good fits are noted across the whole RH range, indicating that the RI coefficients ($m_0$ and $m_1$) derived from the sizing process are accurate. However, it is important to note that we report the RI at 589 nm to be consistent with literature data, and use the Cauchy relation along with $m_0$ and $m_1$ to extrapolate to this wavelength. Our measurements only span the range 640 – 680 nm, and thus we can only be sure of the accuracy

of these coefficients in this wavelength range. The uncertainty range that we quote, ±0.005, reflects this extrapolation, and indeed the RI that we measure appear to be larger than the Lienhard data by ~0.005 (Lienhard et al., 2012). We tested the use of an additional term in the Cauchy expression, $RI(\lambda) = m_0 + \frac{m_1}{\lambda^2} + \frac{m_2}{\lambda^4}$, but observed no significant differences to the RI at 589 nm.

While we acknowledge there is likely some significant uncertainty in the RI at 589 nm from our measurements, another possible source of error comes from the Lienhard data directly (Lienhard et al., 2012), who report an uncertainty of 0.0025 for the RI – a slightly smaller uncertainty than reported here, possibly due to the use of 1st order resonance modes in the Raman spectrum that yield an improved fit. While the exact source of the errors cannot be definitively determined, these observations clearly indicate that the RI is not currently a reliable method of characterizing the

hygroscopic growth due to limited precision and accuracy.

### 3.2.2 Hygroscopicity of 1,2,6-Hexanetriol

The systems explored so far have contained non-volatile or very low volatility solutes. When the vapor pressure is sufficiently high, evaporation of solute from the droplet will occur, leading to a dry size that decreases over time. As we have already identified that using the refractive index to indicate the hygroscopic growth is not accurate, we must

rely on measurements of the size and correct for any changes due to evaporation. 1,2,6-Hexanetriol is an oxygenated organic molecule that has a vapor pressure of 0.2 mPa (Cotterell et al., 2014). Over the course of an hour under dry conditions, a 5000 nm particle will change in size by approximately 50 nm, which leads to a measurable change in the growth curve when using the measured dry size, as shown inset in Figure 6. Evaporation slows as the RH increases and the activity of the evaporating component decreases. By rapidly varying the RH over a humidity cycle, and

updating the dry size each time dry conditions are encountered, the change in dry size due to evaporation can be

corrected, and the growth curve obtained yields excellent agreement to the AIOMFAC prediction, within 1% across the whole measure range. While this system is slowly evaporating, it is likely representative of semi-volatile organic molecules in the atmosphere (Bilde and Pandis, 2001), which will only persist in aerosol if their vapor pressures are sufficiently low.

### 3.2.3 Hygroscopicity of Tetraethylene Glycol

In the case of particles containing solutes with higher vapor pressures than 1,2,6-hexanetriol, measurements must be performed rapidly in order to reduce the amount of evaporation that occurs as the RH scans across the measured range. Tetra-ethylene glycol (tEG) has a vapor pressure of ~16.9 mPa (Krieger et al., 2018), and for a 5000 nm particle will lose around 60 nm of size over 100 s under dry conditions. The evaporation rate is suppressed at higher RH due to a reduction in the activity, and thus significant evaporation occurs only as dry conditions are approached. The size of a droplet exposed to a dehumidifying-humidifying cycle is shown in Figure 7(a), and the corresponding hygroscopic growth of tEG is shown in Figure 7(b). When using the size at the lowest RH to indicate the dry radius, we will overpredict the radial growth factor for the dehumidifying cycle, and under predict the growth factor for the humidifying cycle – in both cases the dry size will be smaller at end of measurement. Even with a RH change over 300 s, this leads to clear upper and lower bounding on the hygroscopic growth curve shown in Figure 7(b). Changing the RH faster is possible, and using a probe droplet to measure the RH avoids the need for fast changing external capacitance probes. However, this will result in a rapid change in size, which will blur the resonance peaks used to determine the size. The speed of the change in RH is thus limited by the minimum exposure time that yields well-resolved peaks of sufficient intensity to characterize. In these experiments, an exposure time down to 0.2 s was used to achieve the 300 s cycle, and this represents the lowest workable timescale with the current experimental configuration. More rapid measurements would be possible with a different spectrometer and illumination source, and these would be necessary to accurately probe the hygroscopicity of material with volatility greater than the compounds reported here.

## 4 Impacts and Applications

Hygroscopic growth measurements are not new, but they continue to play an important supporting role when predicting rates of diffusion, viscosity, phase morphology, optical properties, reactivity etc. The dual-droplet method reported here allows for very accurate measurements of RH and hygroscopicity for a variety of samples over a broad range of conditions. The influence of temperature on hygroscopic growth may be explored without impacting the accuracy of the RH measurements, as the temperature response of the probe droplet can be well-characterized and is positioned *in-situ* with the sample, yielding the best possible accuracy. The high precision in measurements will allow subtle changes in hygroscopicity to be determined, exploring, for example, observations of small changes in diffusion and/or viscosity due to changes in particle composition (Wallace et al., 2021). The accuracy of measurements at high RH will provide an effective bridge between hygroscopic growth measurements under sub-saturated conditions and cloud formation measurements under super-saturated conditions, which can show significant deviations due to solubility and surface tension effects (Petters and Kreidenweis, 2013, 2007, 2008). In addition to characterizing





hygroscopicity, dual-droplet measurements may also be applied to identify the RH and temperature of phase transitions and liquid-liquid phase separation (Stewart et al., 2015; Gorkowski et al., 2020), and RH-dependent water diffusion coefficients (Preston et al., 2017), vapor pressures (Cai et al., 2014), and optical properties (Cotterell et al., 2017; Bain and Preston, 2020; David et al., 2016), in measurements analogous to those previously performed with

single droplets.

## 5 Conclusions

The use of NaCl and LiCl solution droplets as *in-situ* probes of relative humidity has been demonstrated. In both cases, the size of the droplet at the deliquescence point is used to characterize the dry size from the known radial growth factors at these points. This allows the radial growth factor to be determined at any arbitrary RH from measured size.

Using either E-AIM (for NaCl) or a parameterization derived from measurements, we translate the radial GF to RH and show a precision of ~0.5% at 50% and better than 0.1% above 90% RH for NaCl. The LiCl probe was shown to be effective across a wider range of RH owing to its very low deliquescence RH. We used both droplet probes to characterize the RH for a range of measurements to validate the accuracy and applicability of the technique, with excellent agreement shown to established thermodynamic models for ammonium sulfate, citric acid, and 1,2,6-

hexanetriol. We also report the hygroscopic growth of tetra-ethylene glycol, using a rapid RH cycle (over 300 s) to minimize the influence of evaporation on the measured size. These measurements demonstrate the utility and accuracy of the dual-droplet method for accurate hygroscopic growth characterization. Future work will incorporate this approach for characterizing other RH dependent properties, such as water diffusion rates, viscosity, vapor pressures, and optical properties.

**Supplemental Information**

Supplemental information is available here.

**Author Contributions**

JFD designed the research; JC, RK, CS and CP performed the measurements; JFD and JC analyzed the data; JFD and JC wrote the manuscript.

**Competing Interests**

The authors declare that they have no conflict of interest.

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



**Table 1:** Polynomial coefficients describing the RH as a function of radial GF for LiCl and NaCl in low and high RH regimes, using the polynomial: $RH = a + bx + cx^2 + dx^3$ where $x = \frac{GF^3 - 1}{GF^3}$. The GF is determining assuming the deliquescence RH of NaCl is 75.5% with a rGF of 1.90, and the deliquescence RH of LiCl is 11.3% with a rGF of 1.525.

| | LiCl | | NaCl | |
|---|---|---|---|---|
| **Applicable rGF range:** | 1.52 - 1.9 | >1.9 | 1.6 - 2.3 | >2.3 |
| **Applicable RH range:** | 10 - 50% | >50% | 50 - 90% | >90% |
| *a* | 651.9236 | -1607.4685 | 447.7070 | -357.0330 |
| *b* | -1862.2042 | 3280.6543 | -2080.5287 | 800.8240 |
| *c* | 1351.0164 | -1573.3830 | 3070.1008 | -343.8695 |
| *d* | 0 | 0 | -1340.0859 | 0 |



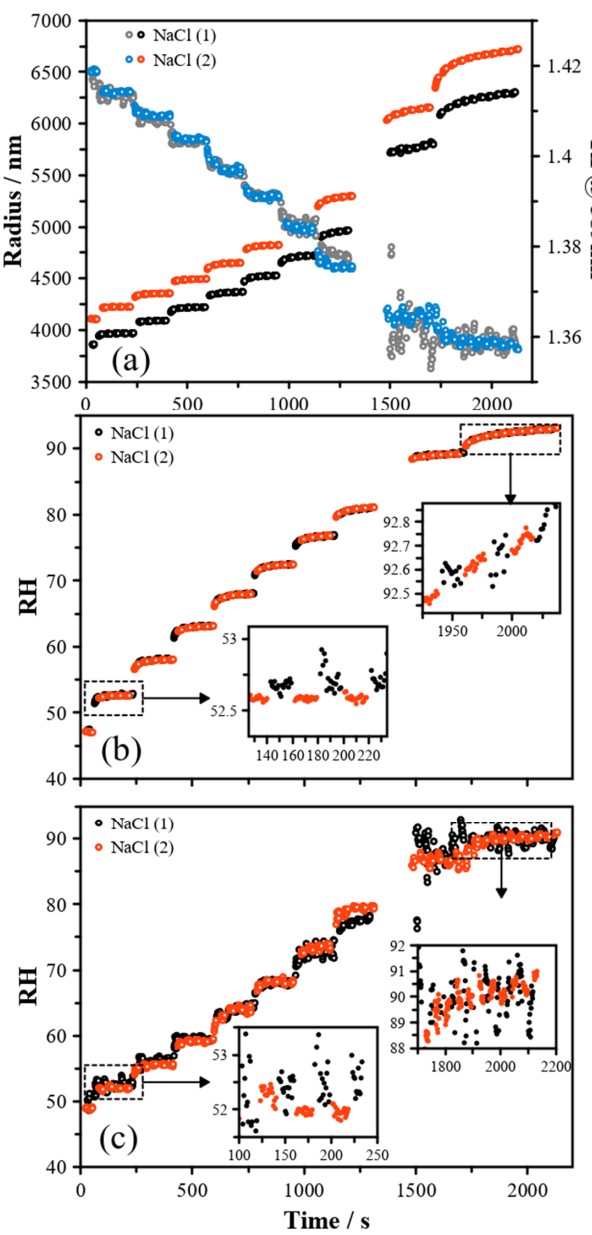

**Figure 1:** (a) The size and RI evolution of two droplets of NaCl held in the LQ-EDB and exposed to a stepwise RH change. (b) Using the measured size at deliquescence, the radial growth factor is obtained and translated to a measure of the RH using a function parameterized to the E-AIM model output (*Table 1*). Agreement in the RH is seen within the estimated uncertainty, which varies from ±0.5% at 50% RH to <±0.1% above 90%. (c) The RI was translated to a RH using the parameterization of Cotterell et al.(Cotterell et al., 2017) yielding a larger uncertainty and a systematic offset from the RH predicted by the size.



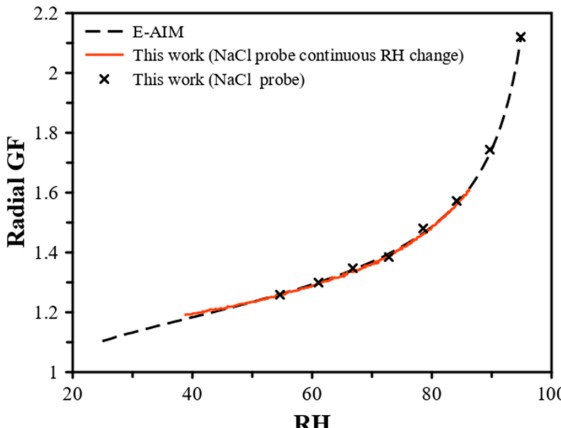

**Figure 2:** Hygroscopic growth of ammonium sulfate determined using a NaCl droplet as a probe of the RH. The points represent a measurement using discrete steps in RH, while the red line indicates a measurement where the RH was changed in 0.5% increments to achieve a pseudo-continuous change in RH. Both datasets agree with the E-AIM prediction of radial growth factor within 1%.



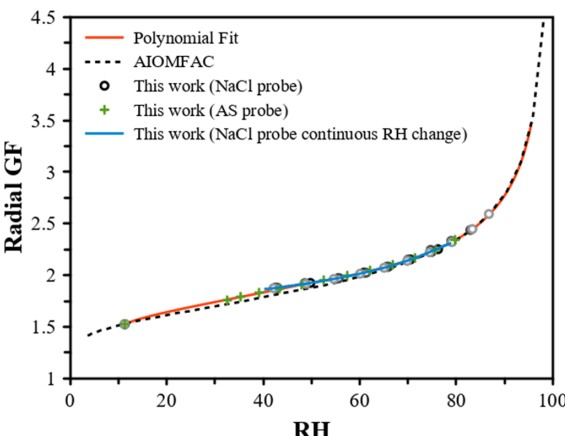

**Figure 3:** Hygroscopic growth of lithium chloride determined using a NaCl droplet as a probe of the RH (black points) and an ammonium sulfate droplet (green points). A continuous RH change measured using an NaCl droplet is shown in blue. The radial growth factor at the deliquescence RH is set at 1.525. The AIOMFAC prediction agrees well at high RH (>70%). A polynomial fit (*Table 1*) was derived to translate measured radial GF of LiCl to the RH for use as a probe under low RH conditions, shown in red.


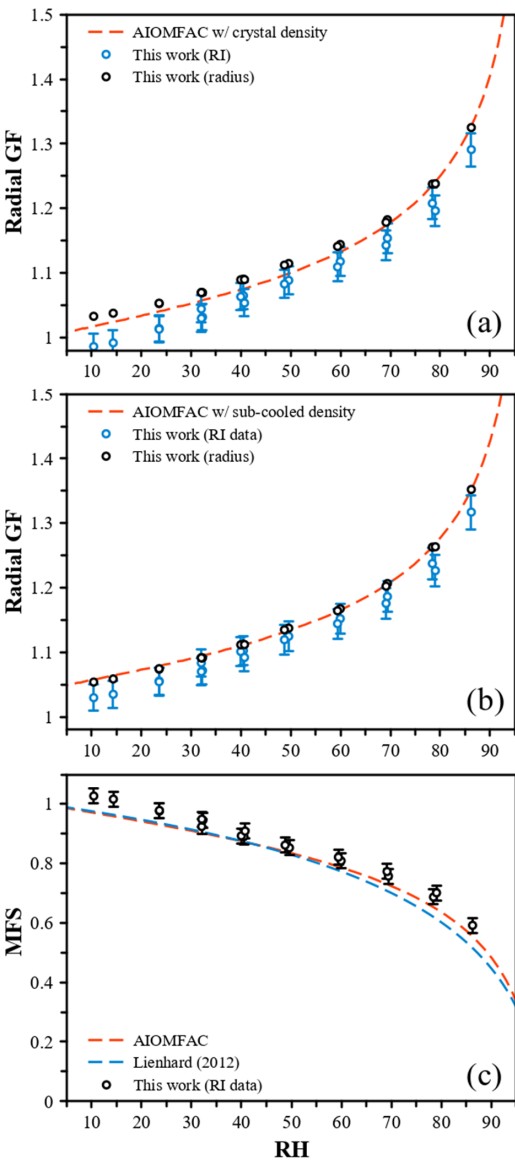

**Figure 4:** (a) The measured radial growth factor (GF) of citric acid was determined from the radius and constrained to match the AIOMFAC prediction at high RH (black points). The radial GF was derived from AIOMFAC assuming a volume-additive density to convert from mass fraction of solute (MFS) to size (red dash line). The radial GF inferred from the measured refractive index (RI), using the parameterization of Lienhard et al. (Lienhard et al., 2012) to convert from RI to MFS, is shown with blue points. The citric acid density was set equal to the crystal density ($\rho =$ 590 $1.66$ g/cm$^3$). (b) The same data at in (A) using a density of citric acid in solution equal to the predicted sub-cooled melt density ($\rho = 1.47$ g/cm$^3$). (c) The MFS derived from the RI (black points), the AIOMFAC model (red-dash line) and the measurements of Lienhard et al. (Lienhard et al., 2012) (blue-dash line).



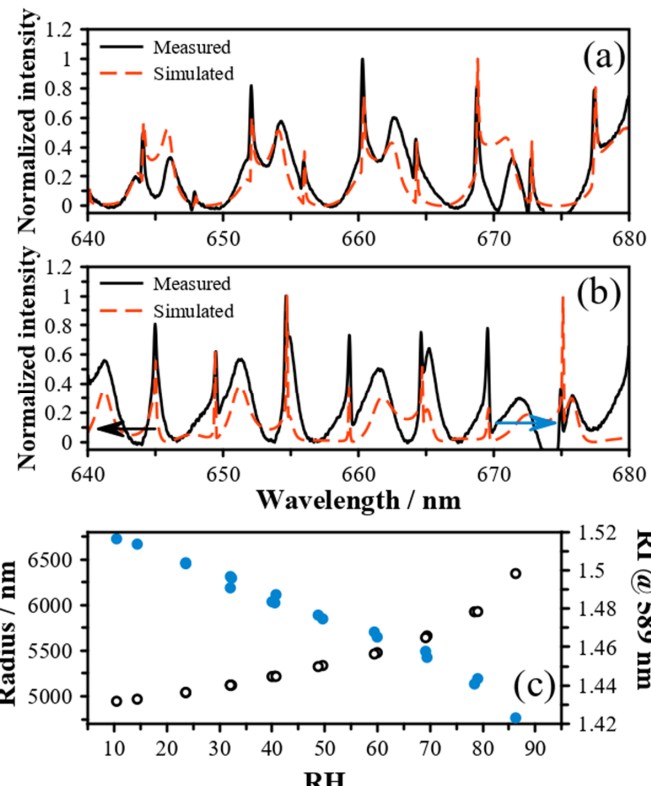

**Figure 5:** Measured spectra for a droplet of citric acid at: (A) high RH (~85%) with a radius of 6328 nm and a RI of 1.4225; (B) low RH (~10%) with a radius of 4954 nm and a RI of 1.5175. Both spectra are shown with the Mie theory simulation using these values as inputs. (C) The size and RI of a citric acid droplet as a function of RH determined using a LiCl probe.



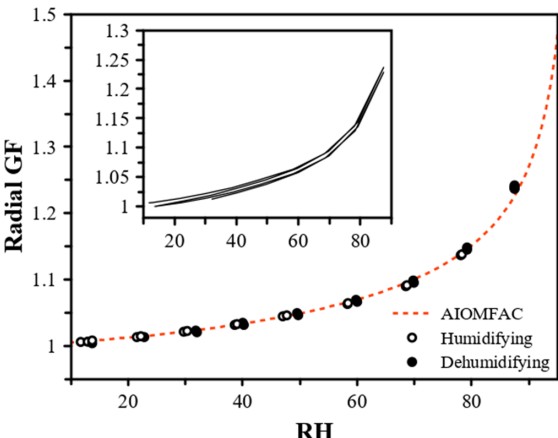


**Figure 6:** Hygroscopic growth of 1,2,6-hexanetriol, an organic molecule with a vapor pressure of $0.20 \pm 0.02$ mPa (Cotterell et al., 2014), leading to evaporation over the course of the measurement. The dry size (at 0% RH was updated on each cycle to achieve agreement to the AIOMFAC prediction (red). A fixed dry size was used in the inset, indicating the influence of a decreasing dry size on the predicted growth factor.


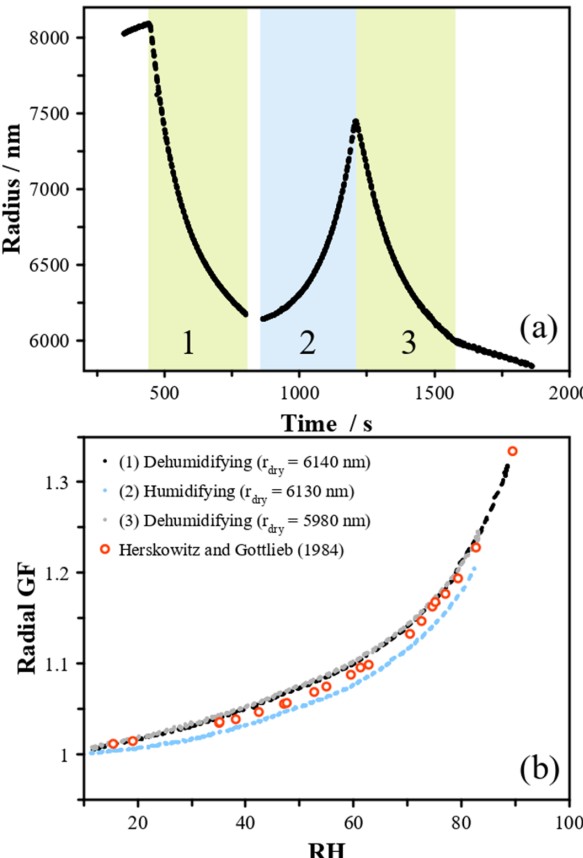

**Figure 7:** (a) Radial evolution of tetraethylene glycol droplet exposed to a rapid pseudo-continuous RH profile. The vapor pressure (~16.9 mPa) results in relatively fast evaporation under low RH conditions, leading to difficulties in defining the dry size. (b) The hygroscopicity was determined with an updated dry size for each humidifying and dehumidifying cycle. The data are compared to the literature measurements of Herskowitz and Gottlieb (using an isopiestic method) (Herskowitz and Gottlieb, 1985; Rard, 2019).

