# Peer review of "Supporting Information: A Dual-Droplet Approach for Measuring the Hygroscopic Growth of Aqueous Aerosol"

_Atmospheric Measurement Techniques, 2021_

## Author Response (AR2)

**Manuscript:** A Dual-Droplet Approach for Measuring the Hygroscopicity of Aqueous Aerosol

**Authors:** Choczynski et al.

We thank all the editors and reviewers for work on this manuscript. All author responses are in blue, and added text is quoted in "quotes".

**Reviewer 1**

The authors presented a linear quadrupole electrodynamic balance (LQ-EDB) method for levitation of dual droplets that can be used for accurate measurement of hygroscopicity of aerosol particles. By using NaCl and LiCl as probe droplets, the range of accurate relative humidity (RH) determination was extended compared to the previous version. The improved technique was applied to study particles containing viscous species and volatile ones, demonstrating its abilities to investigate particles of kinetic limitations and evaporative loss. The experimental setup is well designed and implemented, with great details provided. The experimental results of a few atmospherically relevant species are also clearly presented, backed by literature data and/or thermodynamic model results. The manuscript is generally well written, but I found some places difficult to follow (some examples in technical comments). I therefore recommend publication of this manuscript after Minor Revision. My comments, as shown below, are rather minor too.

We thank the reviewer for their comments and provide a response below.

Section 3.2. It would be helpful to have a schematic or table to show how these three methods to obtain dry size differ, aiding the description here.

We agree – a table (now Table 1, pasted below) has been included to determine the measured variable, the conditions, and the requirements to determine dry size from the data.

|          | Measured value | Conditions | Requirements | Applies to: |
|----------|----------------|------------|--------------|-------------|
| *Method 1* | Size | dRH | Known concentration of saturated aqueous solution | Salts |
| *Method 2* | RI | High RH | Access to calibrated aqueous bulk solution measurements of RI vs conc. | Any aqueous |
| *Method 3* | Size | Dry | Spherical particle under dry conditions | Organic liquids |
| *Method 4* | Size | High RH | Assumes accuracy of thermodynamic model at high RH | Any aqueous |

L142: Why a lower temperature in the chamber than the ambient temperature can avoid condensation in the tubing? Should it be easier to have condensation for a hotter air stream going to a colder region?

We set the temperature of the water bath and the chamber to be slightly lower than ambient (1-2 K). This is a technical detail, and we have found this prevents accumulation of water droplets in the lines that can impact measurements if they are swept into the chamber.  In the supply lines at ambient temperature, the

RH drops thus avoiding condensation. When entering the chamber, the RH increases again to what the levitated particles experience.

Is a standalone Section 5 of Conclusions necessary? Can it be merged into Section 4 as "Conclusions and Implications"?

We agree that these are similar sections, but the Conclusions describes the outcomes of the manuscript while Section 4 describes the broader applications of the method. We consider this suggestion to be stylistic and respectfully opt to retain separation between these sections.

Technical comments:

P3/L72: is it "sample droplet" of "probe droplet" that was used to infer water activity?

Yes, this is now corrected.

P5/L166: "using" to "used"?

Corrected.

P6/L180: what are "these" referring to? Same in L246.

This line have been amended to: "The uncertainty in the size becomes less significant at high RH, while the uncertainty in the RI becomes more significant, due to how the RI varies with RH (Figure S3)."

P6/L186: please delete "Cotterell et al.," in the citation. Same in L210.

Fixed.

P6/L195: ammonium sulfate is indeed one of the most important (and abundant) inorganic components. But it would be inaccurate to say that it is the most abundant salt in the atmosphere, since global estimate suggested that sea salt, which contain mostly sodium chloride, would have a higher burden (a factor of 3 – 4) than that of ammonium sulfate.

Agreed, this has been amended to read: "Ammonium sulfate (AS) is the next most abundant salt in the atmosphere after NaCl and one of the most widely studied atmospherically-relevant inorganic species."

P7/L223: "come" to "comes"?

Corrected.

P8/L241: add "is" before "challenging".

Corrected.

P9/L276: larger than what?

Amended to: "larger than the dried crystalline particle".

P9/L295-296: this sentence is a bit difficult to follow. Maybe better to present it in two sentences.

We have rephrased this portion to read: "While we acknowledge there is likely some significant uncertainty in the RI at 589 nm from our measurements, there is overlap with the literature data when both uncertainty ranges are considered. Lienhard et al. report an uncertainty of 0.0025 for the RI, which is slightly smaller uncertainty than reported here, possibly due to the use of 1st order resonance modes in the Raman spectrum that yield an improved fit."

Figures 2 – 7: it would be good to put a "(%)" after the "RH" in the x-axis title, and use rGF as defined in the text as the y-axis title.

Axis titles have been amended.

**Reviewer 2**

The authors demonstrate a dual-droplet approach for hygroscopicity measurements of micron size aerosol droplets in a linear quadrupole electrodynamic balance. One of the droplets serves a relative humidity sensor; the other is the particle under study. The authors use a spectroscopic method (Mie resonance spectroscopy) for measuring size and refractive index of the particles. They need to switch the two particles alternatively into the measuring volume by adjusting the DC voltage compensating the gravitational force. They illustrate the method by showing several examples of hygroscopicity for different aqueous model systems and show that the accuracy in RH reaches 0.5% at 50% RH and better than 0.1 % at 90% RH. This is significantly better than what can be achieved with traditional RH sensors in electrodynamic balances.

The paper is well written, the topic timely and of interest for the readers of AMT and I recommend publishing but ask the authors to take the following comments and suggestions into account.

We thank the reviewer for their comments and provide a response below.

The main weakness of the presented approach is in my opinion that the technique allows measuring the size growth factor but not directly the mass growth factor. The main advantage of the electrodynamic balance technique is that the inverse of the mass growth factor is the concentration (mass fraction of solute) of the aqueous droplet as long as only water partitions between gas and condensed phase in the experiment. As the authors explain, going from size growth to concentration requires knowledge of the density, which is typically not available for unknown samples. This makes the method less attractive for practical applications where no independent density data exist.

Even more restrictive may be any application to multicomponent particles in which one of the compounds has low solubility and hence the droplet spherical symmetry is no longer conserved. This will make sizing with Mie resonance spectroscopy difficult if not impossible. In their paper describing the setup in detail (Davies, 2019) they make use of the electrostatic force between two particles in the balance to calculate the characteristic constant of the EDB and measure density of a particle by a combination of the optical measurement and the DC voltage to compensate gravitational force. Could this be applied to the two-droplet method here? Please add a discussion.

The method is indeed using radial growth factors. Ideally, we would obtain both radial and mass growth factors, however this is technically challenging with the dual-droplet approach. In our earlier work (Davies 2019 AST), we used the separation of two (more or less) identical droplets in the trap to deduce the trap constant, the droplet charge, and the density, with mass growth factor inherently obtained. With the dual-droplet method described here, the droplets are different, both in terms of size and charge. It is no doubt possible, given the probe droplet's well characterized hygroscopicity response, that it's mass can be inferred, and a total force balance achieved. However, this would require some assumptions on the trap constant (it must be fixed with particle position, or at least well characterized as a function of position), and there must be no vertical air flow. The former may be calibrated, but the latter is a technical challenge than would need to be addressed with a new chamber design that eliminates axial air movement.

We recently published hygroscopic growth data for model lung fluid particles (https://doi.org/10.1039/D1CC00066G) where we report both radial growth and mass growth (via mass fraction of solute). This was achieved by disabling the gas flow altogether for short periods of time to measure the DC voltage in static air, and works well for single droplets, but with inherently less accurate RH data than the dual-droplet approach described here. Thus, there are certainty derivative approaches that would allow accurate mass growth curves to be obtained (such as by calibrating the chamber RH separately with a probe droplet), but we chose to focus solely on the basic case of having two aqueous particles for the purposes of this manuscript.

Minor comments:

In Section 3.1. a more extended discussion on why there is an upper limit in RH measurements because of sizing problems for large particles would be helpful.

To use a probe droplet, we need a known size/RH point, which we chose to be the deliquescence point. We have to reliably size the droplet at this point, and at every other size the droplet adopts as a function of RH. The size range over which the Mie resonance method is reliable is from ~3 to 8 microns. Thus, if at high RH the probe droplet is larger than an upper size, the increased size uncertainty will limit the accuracy of the determined RH.

We have reworded the text slightly, hoping to improve the clarity of explanation: "Despite these advantages, there are some limitations, particularly in the dynamic range of RH that can be probed in a single measurement. Due to the need to measure the size of the particle at dRH in order to determine accurate growth factors, the range of RH that may be probed using LiCl is limited. A particle of 3500 nm at dRH, will be ~8000 nm at 95% RH, and at higher RH will grow to radii that yield increased uncertainty in the size results, translating to increased uncertainty in the RH. For a similar droplet of NaCl, the highest RH that may be reliably determined is ~99%. Less hygroscopic probes may be developed to explore systems closer to saturation, but this range is currently inaccessible to our methods and will not be explored presently."

Lines 265-274: Lienhard et al. (2012) provide experimental density data and refractive index data for citric acid. May be you could compare with those? Fig. 4 would provide an opportunity.

The results reported by Lienhard et al. reference RI and density as a function of mass fraction of solute (mfs). Because our results are tagged to the AIOMFAC output (to derive rGF), any comparison to Lienhard would implicitly also contain a comparison to AIOMFAC, making the discussion overly complicated with little insight gained. In Figure 4(c) we compare the msf vs RH derived from the RI data, and it is clear there are differences between AIOMFAC and Lienhard. To make clearer exactly what the procedure involves, we have revised some of the text on page 8 to allow for a better discussion and comparison:

"We measured the size of citric acid droplets across the full RH range alongside an LiCl droplet to probe the RH. At low RH, measurements were performed for extended time periods (up to hours) to ensure the sample droplet achieved an equilibrium composition. The AIOMFAC-derived growth factor was determined using a volume-additive density approach (see *Supplementary Information*). Briefly, at the highest measured RH of 85%, we estimate from AIOMFAC a MFS of 0.569, which yields 1.29 g/cm$^3$ using the density of the crystal (1.66 g/cm$^3$), giving a dry size for this dataset of 4789 nm and allowing rGF to be determined (Figure 4(a)). If instead we assume the density of the citric acid in solution reflects the sub-cooled density, as predicted using the online tool UManSysProp ($\rho = 1.47$ g/cm$^3$) (Barley et al., 2013), we estimate the density at 85% RH to be 1.22 g/cm$^3$, giving a dry size of 4693nm (Figure 4b). These different dry sizes account for the differences in the experiment results between Figure 4(a) and 4(b). Using

the volume-additive density approach with crystal density shown in Figure 4(a), the AIOMFAC-derived growth curve deviates significantly from our data below ~70% RH. When the sub-cooled melt density is used for the density of CA in solution, the AIOMFAC prediction and data agree closely, with the largest deviation of 0.4% of the radial growth factor (Figure 4(b)). Both predictions reference a dry particle with a density equal to the crystal density. It is clear these data do not tend towards unity under dry conditions and this is due to the formation of a stable hydrate and the amorphous state of the particle – the citric acid will not form a crystal even at 0% RH and thus the size will always be larger than the dried crystalline particle. We further compare our growth curves to the prediction of Lienhard et al., converting their reported solution density to a rGF referencing a crystalline dry particle. This prediction is shown in Figure 4(a) and 4(b), with some deviations from our measurements at high and low RH, respectively, for the two density methods. There is no single value of the dry size that would allow our experimental results to yield agreement to the data of Lienhard et al."

[Figure]

Line 339: How well is the temperature dependence of water activity really known for the reference systems you have in mind? For NaCl there are data available, but the situation is probably not as good for LiCl, correct? For example, Lienhard et al. (2015) reported water diffusivities at temperatures as low as 210 K.

Experiments at higher temperatures 288K to 333K show little variation in aw for NaCl across the range of subsaturated concentrations (e.g. https://onlinelibrary.wiley.com/doi/pdf/10.1111/j.1365-2621.1984.tb12827.x). The range of temperatures over which NaCl water activity is characterized is quite broad, with the E-AIM model usable down to around 250K based on the reference data upon which the model is based. We do not have an estimate of the uncertainty. The deliquescence RH does show some

temperature dependence (e.g. https://pubs.acs.org/doi/10.1021/jp405896y) that would have to be taken into account. We have added some more details to the Impacts and Applications section to reflect this discussion.

For LiCl, the uncertainty is likely much larger, and we would not use LiCl as a probe under non-ambient temperatures. Water diffusivity at low temperatures would yield slower equilibration time, but would not necessarily impact the equilibrium state.

Technical comments:

Line 322: Typo TEG

We have redefined tetraethylene glycol to 4-EG.

Fig. 2 and Fig.3: May be you add a residual panel (E-AIM and the continuous NaCl probe data) to illustrate the agreement.

We have included this for Figure 2 and added a description in the text.

[Figure]

Fig. 5: I recommend using a solid line for the simulated spectra instead of the dashed one; it makes it easier to read. Typo in figure caption: use (a) instead of (A) etc.

We have changed the figure and corrected the caption.

Upon looking again on Fig. 5 of the manuscript (see stext line 285), I would like to ask the authors to comemnt on the significant differences in scatteretd intensity between measured and simulated spectra, e.g. at 670 nm in panel (a). While I agree that resonance positions seem to agree reasonable well this is definetly not true for scattered intensity.

The intensity we measure arises from light that is reflected and scattered by the droplet. We correct for background intensity, but have found that correcting for the reflection is more difficult. The intensity profile of the LED is narrow with steep rising and falling edges, and this leads to some significant issues when correcting the intensity profile to eliminate the reflection contribution. The simulation reports only the scattered light intensity. Given we arrive at size and RI based on peak position alone, there is some

uncertainty (+/- 5 nm in size and ~0.005 in RI). Across this uncertainty range, there is more significant intensity variation than peak position variation. Additionally, the finite resolution of the spectrometer may make sharp peaks appears slightly broader than the simulated counterpart, making the intensity comparison more difficulty. For these reasons, while peak position agreement is good and the overall spectra look comparable, we never expect these data to show perfect agreement. In measurements across a wider wavelength range with a broader illumination spectrum, agreement of both peak positions and overall intensity is more easily achieved (e.g. figure below for an aqueous NaCl particle).

[Figure]

Please explain the clack and Blue arrow in (b).

The arrow in Figure 5B should be located in panel C to indicate that the data in blue corresponds to the RI axis - this will be corrected in the final manuscript.